# Community-Based Efforts Aim to Improve the Food Environment within a Highly Obese Rural Appalachian County

**DOI:** 10.3390/nu13072200

**Published:** 2021-06-26

**Authors:** Rachel Gillespie, Emily DeWitt, Heather Norman-Burgdolf, Brynnan Dunnaway, Alison Gustafson

**Affiliations:** 1Family Consumer Sciences Extension, University of Kentucky, Lexington, KY 40506, USA; emily.dewitt@uky.edu; 2Department of Dietetics and Human Nutrition, University of Kentucky, Lexington, KY 40506, USA; heather.norman@uky.edu (H.N.-B.); brynnan.jacobs@uky.edu (B.D.); alison.gustafson@uky.edu (A.G.)

**Keywords:** rural, food environment, obesity, marketing, behavioral nudge

## Abstract

Rural communities in Appalachia are displaying increased obesity prevalence, yet traditional interventions have not provided a broad enough impact to improve dietary consumption patterns. Therefore, expanding efforts that address the food environment and incorporate behavioral nudges through community-developed marketing strategies may be a viable mechanism to improve food and beverage choices within this unique population. This study installed shelf-wobblers across *n* = 5 gas stations in one rural Appalachian county in Kentucky. Smart Snacks were identified from store inventory lists utilizing the CDC *Food Service Guideline for Federal Facilities* calculator and were categorized into high-protein snacks, low-fat carbohydrate snacks, meal replacement snacks, and no-calorie beverages. NEMS-CS audits were conducted, and monthly sales data was collected at baseline and for six months thereafter for each store location. A difference-in-difference model was used, adjusting for total sales or total mean sales for each Smart Snack model to assess the percentage change within and between stores. Overall, percent change in mean sales and total sales across all stores resulted in a percentage increase of sales of Smart Snack items following wobbler installment. This study provides unique insight into how a community-driven approach to marketing can influence the sale of healthier food and beverage items.

## 1. Introduction

Research suggests that the food environment, including access and availability, influences dietary behaviors among individuals in rural communities [1]. The rural food environment, structurally and economically, is vastly different relative to urban or suburban settings and continues to shift, influencing food access in communities across the United States (U.S.). Individuals residing in rural areas often experience limited access to large supermarkets, while convenience stores and other small stores are readily available [2,3]. The shifting retail food environment in rural communities may be attributed to the rise of discount stores and the closure of local groceries [4]. Additionally, healthy food and drink options may be scarce within food outlets, while calorically dense foods and sugar-sweetened beverages are abundantly available [2]. Consumption of these calorically dense items can lead to increased prevalence of obesity and other diet-related chronic diseases, such as diabetes and heart disease [5]. For this reason, there is great interest in exploring mechanisms to enhance the food environment and improve access to affordable healthy options, ultimately improving diet choice and health outcomes in rural communities.

Evidence consistently reveals the increased prevalence of obesity in rural communities compared to urban and suburban areas [6,7], which can be attributed to a multitude of factors, including socioeconomic status and built environment [7]. At the community level, the food environment shapes purchasing behaviors, which has been linked to dietary intake [8]. Those of a lower socioeconomic status may prioritize price over nutritional quality when making food choices. Thus, food environment modifications that consider product prioritization in rural areas are key. However, much of the existing literature addressing the retail food environment focuses on urban communities, providing few impactful solutions for rural areas that face unique barriers to accessing affordable, nutritious foods [9,10]. This is problematic because populations residing in rural, geographically isolated areas, such as the Appalachian region of the U.S., experience an aberrated food environment relative to those in other rural, urban, and suburban areas [11].

The Appalachian region of the U.S. faces a vast array of barriers hindering the extent of ease to choosing and consuming healthy foods. Spanning 13 states from New York to Mississippi, one quarter of the counties that make up this region are classified as rural, the majority of which are found in eastern Kentucky, where the current study takes place [12]. Research reveals the deep-rooted connection between the scope of specific food outlets—traditional versus nontraditional—and population health [9], thus supporting the idea that greater obesity prevalence may be influenced by the rural food environment [13]. Additionally, all 54 counties located in the Appalachian region of Kentucky are classified as rural [14] and most are economically distressed [15], thereby limiting the variety and depth of accessibility of healthy food items in these communities. The disconnect between demand and availability can often be masked by the persistent poverty many rural communities are currently experiencing within Appalachia. This captures the pervasive dimension of how the longevity of impoverishment affects these communities. Economic distress, the minimized scale of existing food outlets and resources, geographic isolation, and extreme rurality of the region hinders any stimulation and expansion of the food environment.

The economic disadvantage these counties face in Kentucky, relative to other states with higher rural density, has led to nontraditional food venues, such as gas stations, to be economically viable food outlets due to the geographic topography [16]. It is likely that individuals rely on nontraditional, small food stores as a primary food source. Within rural food environments, small food retailers have expressed a willingness to offer healthy options [17], yet they often do not, for numerous reasons. Even when healthy choices are available at these venues, there is less variety and options are more expensive [2]. Therefore, population-level approaches to food system interventions should include environmental approaches to encourage healthier food choices and shopping behaviors as a mode of improving obesity and health status of these communities.

The connection between food environments, food accessibility, and food item purchasing has been well-documented [18,19,20]. A popular approach to improving a food environment within a retail location is by intervening with voluntary strategies to promote healthier food and beverage choices through product placement. Additionally, coupling this intervention with direct acknowledgement of healthier food choices through in-store marketing strategies can facilitate choice architecture more effectively [21]. In stores with unhealthy in-store advertisements, consumers are less likely to perceive the food environment as healthy [22]. Thus, marketing strategies that draw attention to healthier choices may act as a facilitator for improved dietary outcomes. Installing in-store marketing, in the form of shelf-wobblers to identify and draw attention to healthy choices, acts as a behavioral nudge to influence consumer choice, serving as one of the four Ps of marketing—promotion [23]. In other rural communities, behavioral nudges in this form have successfully led to increased recognition of healthy choices and an increase in sales of healthier foods [24]. These strategies present two modes to direct consumers towards specific food and beverage choices by (1) conspicuous subconscious nudging through altered product placement and (2) explicitly indicating healthy food choices for consumers. These strategies create a marketing-mix choice architecture method within food retail settings, which thereby reduces the cognitive toll consumers may experience when shopping and can facilitate healthier purchases. This approach has been suggested as an important and effective tool across various retail food outlets in rural communities [25,26] and should be considered when expanding this intervention strategy to other food outlet settings. Given that eight out of ten rural residents shop for food at gas stations [2,16,27], it is critical to consider how to promote positive behavior change within this venue in the rural food environment. The current rural food environmental landscape is broken, depleted, and bleak, further reinforcing the need for creative strategies to direct consumers toward healthy food and beverage items that are currently available across locally frequented food outlets.

Environmental enhancements tailored to encourage behavior change and healthier purchases must be unique and community-specific. Current literature has explored the effectiveness of obesity-related behavioral interventions promoting healthy choices at stores in various settings. Simple, straightforward shelf placements in low-income urban groceries have been successful at encouraging healthier beverage choices [28], and in-store promotional marketing continues to show promise for influencing healthier shopping behaviors in food retail outlets [29]. Therefore, these mechanisms and strategies are viable interventions, making it important to continue studying their efficacy, particularly in rural community settings, as those that perceive it easier to identify healthy choices are more likely to purchase these items [22]. Examining interventions in rural settings, particularly addressing food environment enhancements, is vital to inform obesity prevention interventions. However, interventions aiming to improve resources, variety, and quality of food items in persistently impoverished rural communities are lacking, which may contribute to poor dietary intake and higher rates of obesity relative to those in urban counterparts.

The aims of this manuscript are to (1) assess the efficacy of in-store promotional marketing targeted to encourage healthier food and beverage items across five gas stations and (2) to measure consumer shopping behavior as a result of “Smart Snack” promotional marketing in a rural Appalachian county. To our knowledge, this is the first study of this kind exploring small store interventions in the rural Appalachian region. These findings may lead to future work improving the dietary choices among populations residing in rural and isolated communities, such as this Appalachian county, which ultimately could improve health and obesity status in areas exhibiting higher prevalence of chronic disease.

## 2. Materials and Methods

### 2.1. Study Setting

This study design is a quasi-experimental time series and is part of a five-year Centers for Disease Control (CDC, Atlanta, GA, USA) High Obesity Program (HOP) project by way of the University of Kentucky Cooperative Extension Service and research study team. These projects support communities in which the adult obesity rate is >40%. The current obesity prevalence in this study-specific Appalachian county in Kentucky is 65% [30], compared to the national obesity prevalence average of 41.4% [31]. Through Policy, Systems, and Environmental (PSE) strategies, sustainable interventions to combat obesity prevalence are designed and implemented with community support, encouraging improved fruit and vegetable intake and increased physical activity [32].

Martin County is the second most eastern county in Kentucky, located in the Appalachian region of the U.S. The county has experienced poverty, high unemployment, and outmigration in the last 10 years [33]. It exhibits persistent poverty, which the United States Department of Agriculture (USDA, Washington, DC, USA) defines as counties with at least 20% population poverty density over the last 40 years [34]. In conjunction with a poverty rate (34.4%) more than triple the national average (10.5%) [33], food insecurity is prevalent in the population, currently resting at 19.2% and projected to increase by nearly 5% in 2021 as a result of the ongoing coronavirus pandemic (COVID-19) [35]. Additionally, more than one fifth (21.6%) of the county’s residents are currently enrolled in the Supplemental Nutrition Assistance Program (SNAP) [36], indicating a strong reliance on food assistance programs for nutritional support in this community.

### 2.2. NEMS-CS Audits

Five gas stations that accepted SNAP EBT cards were identified as partners for HOP activities in Fall 2019. A Nutrition Environment Measure Survey-Corner Store (NEMS-CS) audit was conducted to assess the food environment of each gas station in either the Spring or Fall of 2020 prior to placement of all marketing materials. The NEMS-CS is a validated tool [37] used to assess availability and pricing of a variety of nutritious food choices, including fresh, frozen, and canned fruits and vegetables, compared to less healthy options in eight categories. It also assesses the capacity of store space designed for food (low <25%, moderate 25–50%, most >50%). The NEMS-CS scoring protocol was used to compile a score for availability and price of food choices currently offered at the designated food outlets available to consumers in the county, and an overall score for each gas station was calculated by adding availability and price scores together. This process was repeated for each gas station.

### 2.3. Food Service Guideline Calculator Tool

The “Food Service Guidelines Calculator for Packaged Snacks and Beverages” was developed by CDC [38] to support HOP recipients in determining whether specific snacks and beverages met standards in the *Food Service Guidelines for Federal Facilities* [39]. This calculator aims to provide specific standards for food and nutrition items to encourage the availability, promotion, and consumption of healthier food and beverage items and was therefore pilot tested for use within this food outlet setting [38]. Inventory lists of stocked food and beverage items for the local gas stations (*n* = 5) were provided to the research staff. All food and beverage items from inventory lists (*n* = 1255) were assessed by research staff and one graduate student. Criteria for single-service snack items, multi-serving snack items, and beverages were pre-established in the calculator tool to determine healthfulness of items upon entry of each inventory item. Standards for snack and beverage items included threshold amounts for calories, fat, sodium, and sugar, and snack items designating the first ingredient as a fruit, vegetable, dairy product, or protein food, being a whole grain-rich product, or being a combination food that contains at least ¼ cup of fruit and/or vegetable. All inventory items were inputted into the calculator to compile a master list of designated items for healthy marketing purposes deemed “Smart Snacks” (*n* = 66), which consisted of only 5.26% of the inventory items offered. While the CDC FSG tool provided parameters for marketing placement for this study, the items included in the master list were ultimately at the discretion of research staff. Professional judgement was executed to include some food items not meeting every criterion. For example, nuts exceeded the 200 calorie guideline but were still included as a healthier snack choice in these venues.

### 2.4. Defining and Developing Marketing Tools

Once a master list of Smart Snack items was compiled, the items were separated into four categories: high-protein snack, low-fat carbohydrate snack, meal replacement or protein bar snack, and no-calorie beverages. The nutrition information was gathered from each individual item based on the nutrition facts label. The study team utilized the Google search engine and product information from inventory lists (i.e., product weight, number of servings per package) to determine the nutrition facts for each Smart Snack item. Categorization of snack and beverage items were guided primarily by the first ingredient listed on the nutrition label. High-protein snack items (*n* = 19) qualified if snack foods listed a protein-dense food as the first ingredient, including nuts, seeds, or meat jerky. Low-fat carbohydrate items (*n* = 23) included snacks with a primary ingredient of enriched flour, whole wheat flour, or potatoes, with <4.5 g saturated fat to qualify. Meal replacement and protein bars (*n* = 12) included items that listed multiple protein food ingredients on the nutrition label, such as nuts, seeds, tuna, chicken, whey protein, or whole wheat flour. Lastly, no-calorie beverages (*n* = 12) included beverages that reported zero calories on the nutrition label, including plain or flavored water. The full list of Smart Snack items can be found in Appendix A. Baseline mean sales and total sales were calculated from sales data collected for each of the five gas stations prior to installation of marketing materials.

### 2.5. Timing and Installation of Marketing Materials

Marketing materials were designed and developed as a result of the HOP initiative by the study team in conjunction with the county Health Coalition, comprised of community residents, key stakeholders, and county elected officials. The final Smart Snack materials included the Health Coalition graphic representation, depicting the county outline and mountains in the background to pay homage to the region. The slogan “nourish your body fuel your life” was chosen to discretely encourage nutritious choices, and with guidance from the Health Coalition, figures of individuals were outlined to reflect how the majority of community residents would physically identify themselves. The final design was approved by the Health Coalition prior to installation and can be seen in Figure 1. After categorizing the items into the four categories, the Project Director and one graduate student traveled to all five gas stations to install the marketing materials for each individual Smart Snack item. Placement of Smart Snack shelf-wobblers and in-store marketing took place in October 2020. The graduate student returned in November 2020 to assess placement and determine if any new marketing materials needed to be installed.

### 2.6. Sales Data Capture

Sales data were collected for six individual months (August 2020–January 2021) for each store and then categorized into the corresponding Smart Snack categories. Baseline sales data of Smart Snack items were collected prior to shelf-wobbler installment for the months of August, September, and October. Sales data for the months of November, December, and January were collected following shelf-wobbler installment to assess changes in purchasing over time. The six time points were denoted, and total sales and mean sales for each of the Smart Snack categories corresponding to each store location were documented. Sales data were collapsed for each individual category across all stores and then for each individual store for each month.

### 2.7. Statistical Analysis

Baseline characteristics of the stores and the counties were derived from the NEMS-CS audits and Census data, respectively. To analyze the percent change within each store over the six-month time period and between all the stores, a percent change variable was created for each Smart Snack category for total sales and for mean sales. A difference-in-difference model was used, adjusting for total sales or total mean sales for each model to assess the percentage change within and between stores. Stata SE 16.0 (Stata Corp, LLC, College Station, TX, USA) was used for all analyses.

## 3. Results

### 3.1. NEMS-CS Audits

The NEMS-CS audits were completed across the five gas stations and revealed the following mean scores for each environmental categories: availability 6.2 (scale 0–37), pricing 2 (−9–18), and quality 0 (0–6), with an overall mean total score of 8.2 (range −9–61). The availability scores for the five gas stations, A–E, were as follows: 5, 4, 8, 8, 6, indicating a food environment and product availability poorly conducive to making healthy food and beverage choices. The pricing scores for the gas stations A–E were as follows, corresponding to the availability scores for each store location as reported above: 3, 3, 0, 1, 3. Quality scores of all five gas stations were consistent with a score of zero for each location, due to the lack of availability of fresh fruits or vegetables in each gas station.

### 3.2. Overall Sales

Table 1 depicts the descriptive characteristics of Martin County and pertinent store characteristics for each of the Smart Snack categories. Mean total sales, percentage of Smart Snack purchased to total snacks purchased, and total sales were calculated across six months and all five stores. Mean sales was defined as the average, or measure of central tendency, across a characterized distribution. This was calculated by adding the total sales for each snack category across the six months of collected sales data and dividing by six, indicating the designated time period of distribution. The final value indicates the mean sales for each Smart Snack category. Across the six-month time period, the mean total sales of the 19 high-protein snacks across all stores was $107.00, an increase of 1.97% from preinstallation. The 25 low-fat snack items reflected mean total sales of $45.62 and total overall sales of $1368.68. No-calorie beverages made up the largest percentage of Smart Snack sales (69.9%), while meal replacement food items displayed a 15.05% increase in total sales from baseline. Across all the stores, a change in total sales within the store was observed (−11.06 (−1.98, 1.74)), although individual store locations varied in their total sales performance of Smart Snack items, with three locations (A, B, D) showing an increase in total sales and two locations (C, E) showing a decrease in total sales for Smart Snack items over the six months.

### 3.3. Differences Between and within Stores

Table 2 depicts the percent changes in mean and total sales of Smart Snack items within and between stores across the six months. Looking at mean sales, there was a significant decrease in mean sales for store A for meal replacement snacks and no-calorie beverages. In store B, there was a significant increase in mean sales for low-fat snacks. Store D observed a significant increase in mean sales of high-protein snacks and no-calorie beverages. There were no significant changes in mean sales for stores C or E.

Examining total sales, there was a significant decrease in total sales of meal replacement snacks for store A, while store B had a significant increase in total sales of low-fat snacks. Store D had significant increases in total sales for high-protein snacks and no-calorie beverages; it was the only location with significant increases in more than one category. No significant changes in total sales were observed for stores C or E. However, there was a 1.06% significant increase in total sales across all stores for no-calorie beverages.

Overall, percent change in mean sales and total sales across all five stores resulted in an increased percentage of sales of Smart Snack items. Across all gas stations (*n* = 5) over six months there was a 1.97% increase in mean sales of high-protein snacks, although sales were only statistically significant (21.38 (15.91, 38.68)*) in one store, Store D. Low-fat snacks reflected a 0.55% increase in mean sales across all stores and was statistically significant (44.58 (28.94, 128.12)*) in one store, Store B. Meal replacement snacks showed a 15.05% increase in mean sales across stores and was significant in one store location, Store A (−65.01, (−110.01, −20.04)*). Lastly, no-calorie beverages displayed the greatest increase of 17.23% in mean sales across all store locations, although it was only statistically significant in Store D (61.56 (36.46, 159.29)*). Overall, percent change in mean sales and total sales across all stores resulted in a percentage increase of sales of Smart Snack items following wobbler installment. Individual store locations varied in their total sales performance of Smart Snack items, with three locations showing an increase in total sales and two locations showing a decrease in total sales for items.

## 4. Discussion

This study keenly reveals the bleak food environment within one rural Appalachian county. The NEMS-CS audits verified a food environment poorly conducive to making healthy food and beverage choices prior to any healthy marketing efforts. However, each selected food outlet provided opportunities for enhancement, thereby presenting a medium for community-developed marketing to be installed through this strategy of promotional product marketing. This study is unique in that the promotional marketing implemented was developed through a community-centric lens to create the branding that would be implemented. The Health Coalition has been a guiding body for the community-based participatory research that has taken place, and it has been a foundational objective of the HOP project to accomplish overarching nutrition improvements in communities through local stakeholder relationships. This approach coupled community buy-in with evidence-based research of in-store promotional marketing to facilitate behavior nudging within the context of this study, as community-supported marketing campaigns tend to perform better than outsider-developed initiatives or messaging [41]. This approach is valued particularly in the Appalachian region, in which a grassroots approach to systemic behavior changes is more accepted and often preferred [42].

The promotion of healthy shopping and purchasing choices has been a focal point of public health to reduce obesity and improve health outcomes [24]. Thereby, in-store promotional marketing within this setting provided an opportune outlet to potentially influence shoppers within the community to purchase healthier food and beverage items. By implementing promotional marketing strategies of healthier choices, it targets consumer purchasing habits toward “better-for-you” products in attempt to “nudge” individual behavior. While the remaining P’s within the marketing mix consists of product, place, and price strategies, previous studies have shown that implementation of more than one marketing strategy elicits stronger impact on consumer shopping behavior [24,28]. However, our findings provide evidentiary support for leveraging additional strategies within this geographic and demographic setting for the continuation of these current in-store promotional marketing components.

It is worth exploring why some store locations exhibited certain changes in sales of Smart Snack items compared to other store locations. Figure 2 depicts a map of the store locations included in this study and their respective geographic location in Martin County. Store B is located in the city seat of the county and Store E is in the other official city. Store D, however, in the southeastern most part of the county, performed the best among the five locations, with significant increases in sales in multiple categories. This could be due to surrounding businesses that elicit greater traffic and economic stimulation at the gas station location or due to the close proximity to the neighboring state of West Virginia, which could bring shoppers across state lines to this specific food outlet, prompting an increase in sales compared to the other stores. Store C is configured with an attached fast-food venue, yet it did not exhibit any significant changes in Smart Snack purchases across the six months and rather showed a slight decrease in total sales within the store. Several factors may influence these outcomes, but they reinforce the geographic challenges isolated businesses face within the Appalachian region. Compounded with an impoverished population facing continued outmigration, Stores C and E could depict how lack of economic viability can hinder the behavioral influence of healthier shopping choices at convenience store type food outlets in a rural community.

Additionally, these findings continue to highlight the dire differentiation between urban and rural communities and the subsequent general public health status of their populations. These differences may be due to the layered contributing factors that have come to be largely accepted factors impacting an individual’s health. Socioeconomic status, distance to a food outlet, the number of food outlets available per capita, and availability of healthy and nonhealthy food and beverage items within those food outlets may influence individual behavior [7,43]. Although this study did not collect average travel time, individuals within the Appalachian region have previously reported traveling more than 10 miles one way to grocery shop [44]. Therefore, individuals commonly report shopping at convenience stores or gas stations for home cooking preparations basics, such as eggs and milk, until the next grocery shopping trip [44]. These food access points, such as the locations included in this study, serve as important fixtures within a rural food system, as they provide consistent and periodic exposure points to influence food and beverage purchases.

Future work should consider the expanding scope of food shopping practices, such as through online fronts and platforms. In April 2020, Kentucky joined the USDA pilot program as part of an initiative to encourage online redemption of SNAP assistance dollars through online shopping interfaces with selected grocery stores [45]. Therefore, when considering virtual supermarket venues, salient pricing strategies paired with nudges have successfully led to increased purchase of healthy items, and nudges alone have decreased purchase of unhealthy items [46,47]. This provides a new and pivotal outlet for future work, as what was previously a barrier for SNAP participants now drastically opens a sector of access for this population. Grocery stores have been the predominant focus of expanding accessibility among shoppers of all socioeconomic and geographical residential status, however, this study reinforces the utilization and importance of nontraditional retail food outlets, such as gas stations, for food purchases in rural settings. This is crucial in communities with high SNAP participation, as venues such as these do accept SNAP benefits for certain food items. Therefore, as online grocery shopping continues to expand, choice architecture and healthier marketing landscaping promoting behavioral nudges on a virtual front will be an important consideration for future work as this mode to address food accessibility grows, particularly within this population and rural communities.

This study is not without limitations, as only six months of sales data were collected to assess differences in shopping behavior. However, being part of the broader CDC HOP project provides ample opportunity to continue this work and assess greater consumer shopping patterns over a longer period of time. Additionally, the researchers could not control for the COVID-19 pandemic yet recognize it could have far-reaching effects on purchasing choices. The county included in the study has a projected 20.4% food insecurity rate for 2021 and 9.3% unemployment rate as a result of the onset of the pandemic [35,48], indicating a need for future research within the community. Further, the NEMS-CS audits were collected at different time points, two in one day during Spring 2020 and three during Fall 2020, which could have resulted in remedial differences in audit results. Another limitation could be the “halo effect” [49,50] impacting purchasing behaviors of shoppers, as we do not have individual level purchasing data and only received overall monthly sales data for each of the stores. Future work plans to address and collect individual dietary behaviors to investigate the impact of environmental changes and interventions, as this was not collected as part of this study. This study provides an opportunity for policy-level improvements to determine if marketing strategies influence overall sales of healthier items.

## 5. Conclusions

This study adds to the body of research assessing the efficacy of specific marketing strategies within rural communities aimed to improve food and beverage purchases. Through a local community-centric approach to branding, promotional marketing was developed and installed across five gas stations, promoting healthy food and beverage items to shoppers. This elicited a positive impact on purchasing patterns among shoppers in a rural Appalachian county. This study provides valuable insight into an effective intervention that was channeled through the food environment to improve shopping choices and thus dietary patterns in a vulnerable population that experiences high obesity, barriers to food access, and persistent poverty. Future interventions and work in communities such as these should expand upon these findings by marrying a community-wide marketing campaign to encompass all food outlets with behavioral nudges and other aspects of promotional marketing such as price or positioning alterations.

## Figures and Tables

**Figure 1 nutrients-13-02200-f001:**
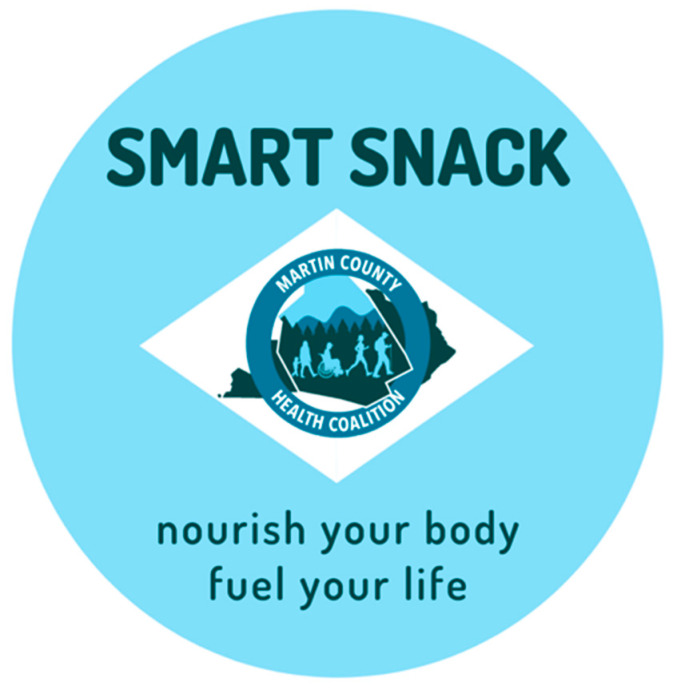
Smart Snack shelf-wobbler placed in gas stations (*n* = 5).

**Figure 2 nutrients-13-02200-f002:**
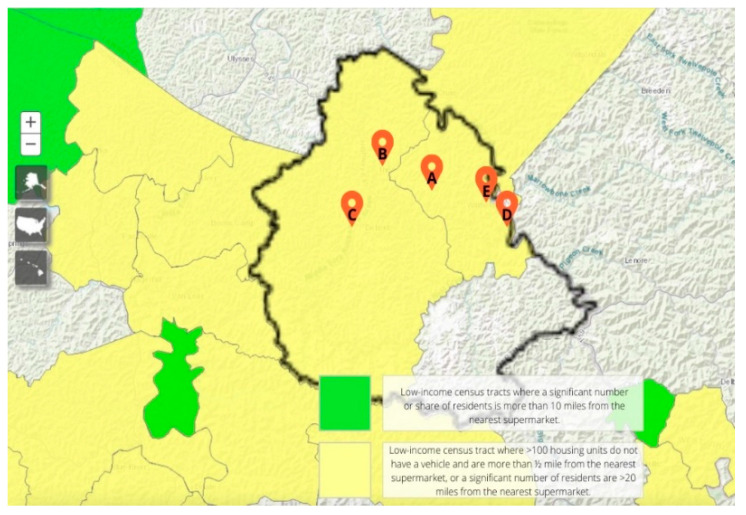
Martin County, KY, USA with designated locations of gas station stores A–E throughout the county.

**Table 1 nutrients-13-02200-t001:** County Level descriptives and store characteristics across all time periods, Martin County, KY, USA.

**County Descriptives**		
Percent Poverty	34.4%	
Percent SNAP Participation	30.7%	
Percent Receiving Free Lunch ^1^	100%	
Median Household Income	$41,013	
**Store Characteristics**		
Mean NEMS-CS Overall Score (*n* = 5)	8.2	
Total Number of High-Protein Snacks	19	
Total Number of Low-Fat Snacks	25	
Total Number of Meal Replacement Snacks	12	
Total Number of No-Calorie Beverages	12	
**Mean Total Sales Across All Stores**		**SE**
High-Protein Snacks	$107.00	10.07
Low-Fat Snacks	$45.62	3.02
Meal Replacement Snacks	$7.05	1.73
No-Calorie Beverages	$447.00	21.08
**Percentage of Smart Snacks/Total Smart Snacks Purchased**		
High-Protein Snacks	16.3%	
Low-Fat Snacks	7.1%	
Meal Replacement Snacks	6.7%	
No-Calorie Beverages	69.9%	
**Total Sales (*n* = 5)**		
High-Protein Snacks	$3127.10	
Low-Fat Snacks	$1368.68	
Meal Replacement Snacks	$1278.78	
No-Calorie Beverages	$13,429.60	
All Sales	$19,204.20	
**Change in Total Sales Across All Stores (*n* = 5)**	−11.06 (−1.98, 1.74)	
Store A	1.42 (−2.93, 5.79)	
Store B	1.22 (−3.73, 6.18)	
Store C	−1.00 (−7.36, 5.35)	
Store D	1.79 (−1.20, 4.79)	
Store E	−4.00 (−13.09, 5.84)	

^1^ The Community Eligibility Provision dictates that in school districts with student bodies in which ≥40% qualify for free or reduced lunch as part of the National School Lunch Program (NSLP), the entire school will be enrolled in free and/or reduced lunch [40]. Across all schools (*n* = 5) in Martin County, >40% of the student body in each school qualifies for free lunch, therefore, the entire student body in the county receives free lunch as part of the NSLP. SNAP is defined as Supplemental Nutrition Assistance Program; NEMS-CS is defined as Nutrition Environment Measure Survey-Corner Store.

**Table 2 nutrients-13-02200-t002:** Percent change in mean and total sales over time within and between stores.

	**Percent Change in Mean Sales Store A**	**Percent Change in Mean Sales Store B**	**Percent Change in Mean Sales Store C**	**Percent Change in Mean Sales Store D**	**Percent Change in Mean Sales Store E**	**Percent Change in Mean Sales Across All Stores**
Mean Sales on High-Protein Snacks	−9.25 (−25.04, 6.04)	2.96 (−23.34, 29.31)	10.81 (−26.72, 48.33)	21.38 (15.91, 38.68) *	0.40 (−72.69, 73.43)	1.97 (−10.08, 14.03)
Mean Sales on Low-Fat Snacks	−5.61 (−55.98, 44.76)	44.58 (28.94, 128.12) *	−7.17 (−35.04, 20.03)	4.23 (−29.18, 37.66)	20.02 (−55.48, 95.54)	0.55 (−14.72, 15.83)
Mean Sales on Meal Replacement Snacks	−65.01 (−110.01, −20.04) *	48.92 (−90.59, 188.43)	7.46 (−155.65, 170.59)	61.88 (−136.75, 260.49)	N/A	15.05 (−35.11, 65.21)
Mean Sales on No-Calorie Beverages	−65.39 (−110.61, −20.11) *	32.79 (−98.08, 163.89)	7.27 (−155.46, 170.02)	61.56 (36.46, 159.29) *	−15.67 (−28.98, 33.54)	17.23 (−31.34, 65.80)
	**Percent Change in Total Sales Store A**	**Percent Change in Total Sales Store B**	**Percent Change in Total Sales Store C**	**Percent Change in Total Sales Store D**	**Percent Change in Total Sales Store E**	**Percent Change in Total Sales Across All Stores**
Total Sales on High-Protein Snacks	15.84 (−27.81, 59.49)	5.47 (−13.17, 24.11)	8.93 (−55.06, 72.92)	40.43 (14.20, 95.06) *	33.64 (−80.96, 148.26)	9.60 (−2.18, 21.39)
Total Sales on Low-Fat Snacks	−11.84 (−94.32, 72.04)	5.49 (1.10, 10.65) *	−10.24 (−55.96, 35.84)	4.29 (−38.64, 47.24)	7.95 (−157.99, 173.89)	−0.46 (−14.27, 13.34)
Total Sales on Meal Replacement Snacks	−10.21 (−158.06, −137.65) *	18.13 (−268.21, 304.97)	27.01 (−133.72, 79.02)	0.42 (−0.99, 5.63)	N/A	−6.41 (−53.43, 40.73)
Total Sales on No-Calorie Beverages	−3.56 (−25.53, 18.39)	4.08 (−21.01, 29.18)	0.01 (−18.03, 18.07)	8.56 (4.83, 21.95) *	−2.59 (−23.17, 17.35)	1.06 (0.26, 1.85) *

* Indicates *p* < 0.05.

## Data Availability

The data presented in this study are available on reasonable request from the corresponding author.

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
