# Peer review of "Community-Based Efforts Aim to Improve the Food Environment within a Highly Obese Rural Appalachian County"

_nutrients, 2021, doi:10.3390/nu13072200_

Round 1

Reviewer 1 Report

In the manuscript by Rachel Gillespie et al., the authors demonstrated how a specific marketing within rural communities where the percentage of obese people is very high, can improve shopping choices and, consequently, the dietary models. 

In general, the structure of the proposed manuscript is presented clearly and the analysis is accurate and precise; the tables are instrumental for the content of the paper.

However, the authors should indicate the caloric value of these snacks and investigate whether these snacks are predominantly consumed by obese people. These data could improve knowledge within the scientific community and keep vigilant on obesity, a public health problem

Author Response

Thank you for this unique perspective, however, due to the limitations of the pandemic and study design, we were unable to collect individual-level data and cannot explicitly indicate whether obese individual shoppers consumed the high caloric snacks. The next step is applying our findings to human subjects in an effort to marry environmental and macro-scale interventions with individual behavior changes. Thank you for your valuable feedback and insight.

We added language indicating this as a study limitation at Lines 400-202: “Future work plans to address and collect individual dietary behaviors to investigate the impact of environmental changes and interventions, as this was not collected as part of this study.”

Lines 431-497, Appendix A: We included the caloric content for all Smart Snacks. 

Reviewer 2 Report

The authors describe the effect of in-store promotional marketing targeted to encourage healthier food and beverage items across five gas stations in a rural Appalachian county with a population with a high proportion of obese people.

The study is well done and interesting, helping to increase our knowledge about habits and their change after a promotional marketing approach in a rural community.

The only aspect to consider is what the authors define as total sales and mean sales; both concepts should be defined. The concept of total sales is intuitive and easy to understand (the total income generated from all sales), but the concept of mean sales is not clear what it refers to, so it should be defined and the authors should explain how it is calculated.

Author Response

Thank you for this valuable feedback. We elaborated on the definition and specific calculation that was used to determine mean sales, as you are correct in that mean sales is not as intuitive and simplistic as total sales is. We appreciate this attention to detail and have added the following language to reflect the definition as it applies to this manuscript context. 

Lines 256-259: “Mean sales is defined as the average, or measure of central tendency, across a characterized distribution. This is calculated by adding the total sales for each snack category across the six months of collected sales data, and diving by six, indicating the designated time period of distribution. The final value indicates the mean sales for each Smart Snack category.”